# Clinical Applications of Nano-Hydroxyapatite in Dentistry

**Rossana Izzetti** *⬮, **Stefano Gennai** ⬮, **Marco Nisi, Francesco Gulia, Marco Miceli and Maria Rita Giuca**

Unit of Dentistry and Oral Surgery, Department of Surgical, Medical, Molecular Pathology and Critical Care Medicine, University of Pisa, 56126 Pisa, Italy
* Correspondence: rossana.izzetti@med.unipi.it

**Abstract:** Nano-hydroxyapatite (nano-HAp) is a biomaterial which is finding increasing application in dentistry due to its osteogenic properties and biocompatibility. The aim of the present study is to review the clinical applications of nano-HAp in dentistry. A literature search was performed in PubMeD and Scopus. In total, 154 articles were retrieved. Following title and abstract analysis, 122 articles were excluded. Further 6 articles were excluded through full-text analysis. The systematic review was conducted on 26 articles. In 3 studies, nano-HAp was employed for tissue remineralization, 8 studies applied nano-HAp for the treatment of dentin hypersensitivity, 2 studies evaluated the role of nano-HAp in orthodontics, 9 studies applied nano-HAp as a graft material, and 4 studies applied nano-HAp to periodontology and implantology. The available evidence supports the use of nano-HAp in the prevention of caries, as a desensitizing agent in the treatment of dentin hypersensitivity and as a graft material. It should be, however, highlighted that further research appears mandatory in assessing and validating the full potential of nano-HAp in clinical dentistry.

**Keywords:** nano-hydroxyapatite; hydroxyapatites; dentistry; operative dentistry

## 1. Introduction

Calcium phosphate salts are among the main components of bone and teeth [1]. Calcium apatite accounts for 50% of volume and 70% of weight of human bone [2,3]. Hydroxyapatite (HAp) is a calcium phosphate compound presenting identical characteristics to bone apatite, such as hexagonal structure and a stoichiometric Ca/P ratio of 1.67 [4]. Carbonated calcium-deficient HAp is the main mineral of which dental enamel and dentin are composed [5]. In dental enamel, the inorganic matrix accounts for 96% in weight, with HAp crystallites size ranging between 48–78 nm. In mature dentin, the inorganic matrix represents the 70% in weight, and HAp crystallites size ranges between 60–70 nm [6].

Compared with other calcium phosphates, HAp is characterized by high thermodynamic stability under physiological conditions [4]. The possibility to chemically synthesize HAp has extended the clinical applications in several medical fields, especially for its osteogenic properties and biocompatibility [7]. HAp synthesis can be performed through various techniques, including dry methods, wet chemical reactions, or mechanochemical reactions, which allow to obtain different compounds in terms of particle size, shape, and chemical composition [1,8].

Among the HAp structures, nano-sized HAp (nano-hydroxyapatite, nano-HAp) is a biomaterial of particular interest due to its similarity in size, crystallography and chemical composition with human hard tissues [9]. While synthesized HAp is characterized by a crystal size of 25.40 nm, nano-HAp crystal size ranges between 20 to 80 nm [9].

Nano-HAp finds several applications in dentistry, due to its characteristics of biocompatibility and bioactivity [10]. Among its properties, the ability to promote re-mineralization appears extremely valuable in presence of early caries, where nano-HAp promotes direct replacement of lost minerals or carries mineral ions to the collagen network [10]. In cases of dentin hypersensitivity, nano-HAp has been reported to induce occlusion of dental tubules

by acting as a mineralizing agent [10]. Moreover, nano-HAp is employed for implant coating as it improves bone to implant contact and promotes bone adhesion, along with having bacteriostatic properties [10]. Such characteristics account for the favourable outcomes of nano-HAp use as graft material, enhancing angiogenesis and bone healing [10].

The aim of the present study is to review the current applications of nano-HAp in dentistry.

## 2. Materials and Methods

### 2.1. Protocol Development and Eligibility Criteria

The review protocol was prepared according to the Preferred Reporting Items Systematic review and Meta-Analyses (PRISMA) [11–13], and registered in PROSPERO (Prospective Register of Systematic Reviews). The following focused question was phrased: "What are the clinical applications of nano-HAp in dentistry?"

The inclusion criteria were:

- Human studies
- Observational, case-control, randomized control ed trial and interventional study design
- A minimum of 10 participants in the study
- Articles written in English

The exclusion criteria were:

- Animal studies
- In vitro studies
- Systematic reviews and review
- Case reports and case series with less than 10 participants
- Articles not written in English.

No time limitations were set.

### 2.2. Information Sources and Search

The electronic search was applied to the Cochrane Oral Health Group specialist trials, MEDLINE via PubMed and Scopus (SG) up to August 2022. The strategy used was a combination of MeSH terms and free text words (in dental journals):

(Nano-hydroxyapatite OR nano-HA) AND (Dentistry OR Dentistry [MeSH Terms] OR Dentistry, Operative [MeSH Terms] OR Dental Surgery OR Oral Surgery OR Oral Surgery [MeSH Terms] OR Oral Surgical Procedures [MeSH Terms]).

Hand search was also performed and bibliographies of all relevant papers and review articles were checked to detect additional studies. Trials databases such as clinicaltrial.gov and other relevant sites were searched.

### 2.3. Study Selection and Data Collection

Eligibility assessment was performed through title and abstract analysis and full text analysis in July 2022. Title and abstract analysis was performed by two calibrated reviewers (RI and MN, κ-score > 0.8) for possible inclusion in the review. In cases of unclear abstracts (e.g., incomplete reporting, unclear study methods), further assessment through full text analysis was performed order not to exclude any potentially relevant articles. For the studies deemed suitable for inclusion following full text analysis, data extraction was carried through an ad hoc extraction sheet.

### 2.4. Risk of Bias

A process of quality analysis performed according to the Cochrane Reviewers' Handbook [14] was carried out by three reviewers (RI, MN, SG) to evaluate the risk of bias. Quality analysis assessed the following domains, which were deemed adequate, inadequate, or unclear:

- Random sequence generation

- Allocation concealment
- Blinding of participants, personnel, and outcome assessors
- Handling of incomplete outcome data
- Selective outcome reporting.

*2.5. Summary Measures and Synthesis of the Results*

Data were synthesized in evidence tables addressing study characteristics and main conclusions.

**3. Results**

*3.1. Study Selection*

Database search retrieved 131 articles in total. Hand search retrieved further 23 papers, for a total of 154 articles. One-hundred and twenty-two articles were excluded following title and abstract analysis. Full text analysis was performed for the remaining 32 articles, and 6 articles were further excluded. Final assessment was carried out on 26 articles (Figure 1).

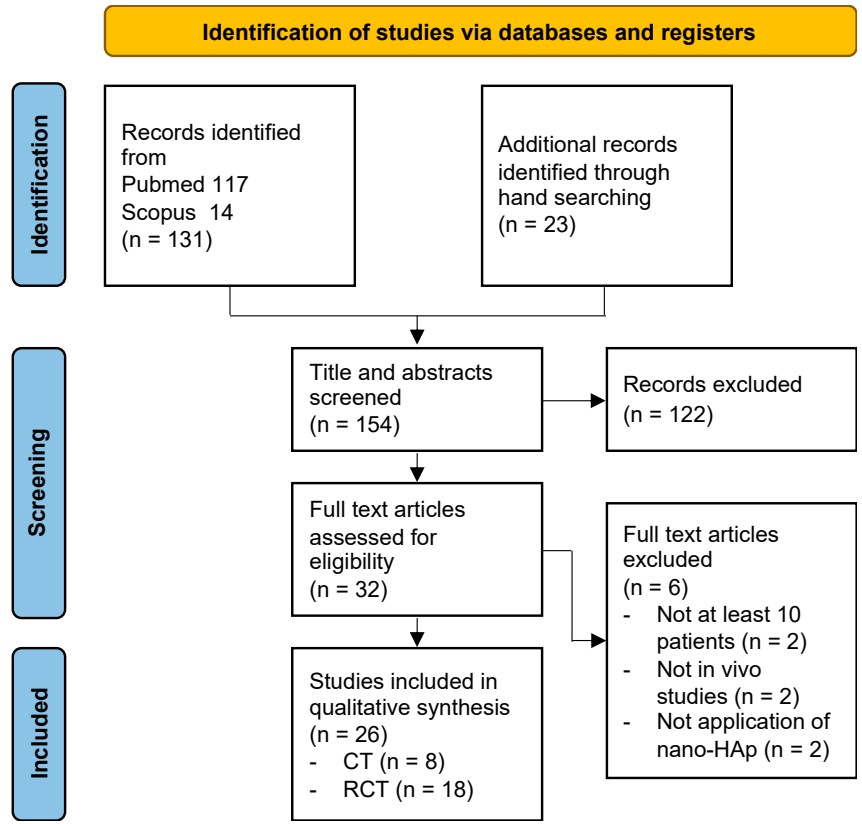

**Figure 1.** Flowchart of the literature search according to PRISMA guidelines.

*3.2. Study Characteristics*

The study population consisted of 899 subjects (51.8% females), with a mean age of 38.49 years (SD ± 12.88). The follow-up period varied from a minimum of 2 weeks to a maximum of 24 months. Three studies investigated the application of nano-HAp in tissue remineralization [15–17]. Eight studies evaluated the application in the reduction of dentin hypersensitivity [18–25]. Two studies evaluated the role of nano-HAp in orthodontics [26,27]. Nine studies applied nano-HAp as a graft material [28–36]. Three studies applied nano-HAp to periodontal defects and one study employed nano-HAp for implant coating [37–39].

### 3.2.1. Tissue Remineralization

Najibfard et al. [15] evaluated the degree of enamel remineralization following exposure to dentifrices containing different concentrations of nano-HAp (5% or 10% nano-HAp) versus 1100 ppm fluoride dentifrice and 10% nano-HAp delivered through an intra-oral appliance. The authors did not identify statistically significant differences between groups in terms of mineral loss and lesion depth. All dentifrice formulations promoted significant mineral gain in the enamel subsurface lesions.

Wierichs et al. [16] compared the effects of four different dentifrices on remineralization. The experimental dentifrices contained a combination of zinc-carbonate and hydroxyapatite nanocrystals, different fluoride concentrations (1100 or 5000 ppm F), and a fluoride free control. The authors found that fluoride free and nano-HAp containing dentifrice did not hamper demineralization, which was reduced on a dose-dependent response for fluoride containing dentifrices.

Grocholewicz et al. [17] evaluated enamel remineralization on initial approximal caries following treatment with a nano-HAp gel, gaseous ozone therapy, or a combination of both. The authors found at one year follow-up a regression of the initial caries lesions in 36.5% of patients treated with nano-HAp gel, in 60% of patients treated with ozone therapy and in 69.3% of patients treated with both. At two years, caries reversal was observed in 18% of cases for the nano-HAp group, 38% of cases in the ozone therapy group, and in 45% of the combined treatment group. The authors concluded that the combination of nano-HAp gel and ozone therapy produced the best effect compared with nano-HAp or ozone therapy alone

Results are summarized in Table 1.

### 3.2.2. Dentin Hypersensitivity

Browning et al. [18] compared the effect of a nano-HAp paste versus placebo in reducing sensitivity following dental bleaching. Placebo group reported sensitivity in 51% of cases, with a mean duration of symptoms of 76 days, and the nano-HAp group showed sensitivity in 29% of cases for a mean of 36 days. The authors concluded that the use of a nano-HAp paste following bleaching significantly reduced the incidence and duration of tooth sensitivity.

Vano et al. [19] evaluated the effectiveness of a bleaching technique with 6% hydrogen peroxide with or without 2% nano-HAp. The adjunct of nano-HAp produced significantly lower sensitivity at 24 h post treatment. No difference was found between treatment groups at 7 and 14 days post-bleaching. The authors concluded that the two techniques had similar outcomes in terms of bleaching, and that the adjunct of nano-HAp reduced tooth sensitivity at 24 h post-treatment.

Wang et al. [20] compared four different desensitizing agents: (i) nano-HAp professional paste (100-nm size 20% HAp, potassium nitrate, NaF, and 9000 ppm F); (ii) nano-HAp professional paste plus a paste containing 100-nm size 10% HAp, potassium nitrate, NaF, and 900 ppm F; (iii) 8% arginine and calcium carbonate paste plus 8% arginine, calcium carbonate, sodium monofluorophosphate, and 1450 ppm F toothpaste; and (iv) NaF professional varnish with 22,600 ppm F. At the first month, no significant differences were found between treatments, which all reduced hypersensitivity. At three months, no significant difference was found among the treatments, although NaF varnish had the lowest effect in reducing hypersensitivity. Treatment with 8% arginine and calcium carbonate paste plus 8% arginine, calcium carbonate, sodium monofluorophosphate, and 1450 ppm F toothpaste increased its effect over time. Nano-HAp was as effective as the other treatments in reducing hypersensitivity.

**Table 1.** Summary of Nano-HAp application in tissue remineralization.

| Author | Setting, Funding | Study Design | Sample Size Number (% of Females), Mean Age, Number of Sites | Follow-Up (Months) | Interventions | Study Conclusions |
|---|---|---|---|---|---|---|
| Najibfard et al. [15] | Dental Hospital, Industry | RCT | 30 patients, (60% F), mean age 37.8 years, 90 sites | 1 | 1. 5% Nano-HAp dentifrice 2. 10% Nano-HAp dentifrice 3. 1100 ppm Fluoride dentifrice | Dentifrices containing nano-HAp showed the same remineralizing capacity as a fluoride dentifrice |
| Wierichs et al. [16] | Dental Hospital, Public Funding | RCT | 20 patients (75% F), mean age 20–56 years, full mouth | 0,75 | 1. Zinc-carbonate Nano-HAp dentifrice 2. 0 ppm Fluoride dentifrice 3. 1100 ppm Fluoride dentifrice 4. 5000 ppm Fluoride dentifrice | A fluoride dose-response on dentin was noted, while fluoride free dentifrice containing Nano-HAp did not hamper demineralization |
| Grocholewicz et al. [17] | Dental Hospital, NR | RCT | 92 patients (71% F), mean age 23.3 years, 546 sites | 24 | 1. Nano-HAp gel 2. Ozone therapy 3. Nano-HAp gel and ozone therapy | Nano-HAp gel and ozone therapy re-mineralized initial approximal enamel and dentine subsurface lesions of premolars and molars |

Abbreviations: RCT, randomized clinical trial; NR, not reported.

**Table 2.** Summary of Nano-HAp application in dentin hypersensitivity.

| Author | Setting, Funding | Study Design | Sample Size Number (% of Females), Mean Age, Number of Sites | Follow-Up (Months) | Interventions | Study Conclusions |
|---|---|---|---|---|---|---|
| Browning et al. [18] | Dental Hospital, NR | RCT | 42 patients (F not reported), mean age not reported, full mouth | 1 | 1. Nano-HAp toothpaste 2. Placebo | Nano-HAp paste following tooth whitening was associated with a significant reduction in the duration of tooth sensitivity |
| Vano et al. [19] | Dental Hospital, Self supported | RCT | 60 patients (55% F), mean age 28 years, full mouth | 9 | 1. 6% hydrogen peroxide with 2% nano-HAp 2. 6% hydrogen peroxide without 2% nano-HAp | 2% nano-HAp resulted in significant lower tooth sensitivity at 24 h post- bleaching. |

Table 2. *Cont.*

| Author | Setting, Funding | Study Design | Sample Size Number (% of Females), Mean Age, Number of Sites | Follow-Up (Months) | Interventions | Study Conclusions |
|---|---|---|---|---|---|---|
| Wang et al. [20] | Dental Hospital, Public Funding | RCT | 28 patients (75% F), age 18–60 years, 137 sites | 5 | 1. Nano-HAp professional paste (100-nm size 20% HAp, potassium nitrate, NaF, and 9000 ppm F) 2. Nano-HAp professional paste plus a paste containing 100-nm size 10% HAp, potassium nitrate, NaF, and 900 ppm F 3. 8% arginine and calcium carbonate paste plus 8% arginine, calcium carbonate, sodium monofluorophosphate, and 1450 ppm F toothpaste 4. NaF professional varnish with 22,600 ppm F | Nano-HAP (with or without home-careproduct association) was as effective as the other treatments for reducing hypersensitivity over three months. |
| Gopinath et al. [21] | Dental Hospital, Self-supported | RCT | 36 patients (44% F), age 18–60 years, full mouth | 1 | 1. Nano-HAP containing toothpaste 2. 5% calcium sodium phosphosilicate toothpaste | Nano-HAp showed significant reduction in dentine hypersensitivity at 4 weeks |
| Bevilacqua et al. [22] | Dental Hospital, NR | RCT | 30 patients (F not reported), age 18–60 years, full mouth | 3 | 1. Fluoride gel 2. Fluoride gel and nanostructured desensitizing agent | All tested desensitizing agents reduced dentin hypersensitivity at 3 months, without significant differences |
| Jena et al. [23] | Dental Hospital, Industry | RCT | 45 patients (62% F), age 18–50 years, 122 sites | 3 | 1. Desensitizing paste containing 5% calcium sodium phosphosilicate (NovaMin) 2. Desensitizing paste containing 8% arginine 3. 15% hydroxyapatite nanoparticles toothpaste | 15% nano-HAp toothpaste was the most effective desensitizing agent followed by 8% arginine and 5% NovaMin group |
| Anand et al. [24] | Dental Hospital, Industry | RCT | 60 patients (58% F), mean age 42.25 years, full mouth | 1 | 1. Nano-HAp toothpaste 2. 8% arginine containing toothpaste | Nano-HAp based and arginine based tooth pastes reduced dentin hypersensitivity |
| Vano et al. [25] | Dental Hospital, Public Funding | RCT | 105 patients (61% F), mean age 42 years, full mouth | 1 | 1. Nano-Hap 2% toothpaste gel fluoride free 2. Fluoride toothpaste 3. Placebo | Nano-HAp gel dentifrice was effectivein reducing dentin hypersensitivity |

Abbreviations: CT, controlled trial; RCT, randomized clinical trial; NR, not reported.

Gopinath et al. [21] evaluated the desensitizing effects of Nano-HAp toothpaste versus 5% calcium sodium phosphosilicate. Both treatments provided a significant reduction in sensitivity, as measured through Visual Analogue Scale (VAS), tactile test, air blast, and cold water, over the 4 weeks observation period. No statistically significant differences were noted between the two study products in the reduction of sensitivity from baseline to 4 weeks, for any of the sensitivity parameters.

Bevilacqua et al. [22] compared the effects of fluoride gel with or without a nanostructured desensitizing agent. The authors did not find significant differences between treatments, which both provided a decrease in mean pain intensity over the 3 months period of observation.

Jena et al. [23] evaluated the effectiveness of three desensitizing toothpastes, containing 5% calcium sodium phosphosilicate (NovaMin), 8% arginine, and 15% hydroxyapatite nanoparticles, respectively. A reduction of dentin hypersensitivity in terms of VAS score and Schiff Cold Air Sensitivity Scale was observed in all groups at all timepoints. The most effective desensitizing agent was 15% nano-HAp containing toothpaste, followed by 8% arginine and 5% NovaMin group.

Anand et al. [24] compared nano-HAp versus 8% arginine toothpaste in reducing dentin hypersensitivity. Both toothpastes were equally effective in reducing dentin hypersensitivity in the absence of statistically significant differences.

Vano et al. [25] compared a nano-HAp 2% gel toothpaste, a fluoride gel toothpaste, and a placebo in reducing dentin hypersensitivity. Nano-HAp treatment resulted in lower sensitivity values at two and four weeks follow-up.

Results are summarized in Table 2.

### 3.2.3. Orthodontics

The study by Malekpour et al. [26] investigated the effect of nano-HAp serum in adjunct to finishing and polishing techniques (carbide burs and Sof-Lex discs) on enamel colour alterations following debonding procedures in orthodontic treatment. Nano-HAp did not produce significant improvement in reducing tooth colour changes after debonding.

Enan & Hammad [27] compared the degree of microleakage under orthodontic bands using luting glass-ionomer cement with different concentrations (0%, 5%, 10%, 15% by weight) of nano-HAp. Significant positive effect of nano-HAp was observed in reducing microleakage around orthodontic bands, especially in cases of 15% nano-HAp enriched glass-ionomer cement.

Results are summarized in Table 3.

### 3.2.4. Bone Grafts

Strietzel et al. [28] employed nano-crystalline HAp for lateral ridge augmentation. The authors observed a significant gain in alveolar ridge width, and no implant failure was registered at 24 months follow-up. It was concluded that lateral augmentation with nano-HAp provided a quantitatively and qualitatively sufficient site for primary stable implant placement.

Canullo & Sisti [29] evaluated nano-structured Mg-enriched HAp covered with covered with a titanium reinforced extended polytetrafluoroethylene membrane for vertical ridge augmentation. Implants were positioned contextually and loaded two weeks later. The authors did not observe implant loss, and inter-implant regenerated bone levels remained stable at 12 and 24 months.

Nano-structured Mg-enriched HAp was also used for socket preservation by Canullo et al. [30], who found complete regeneration of the alveolar socket at 12 months, with mineralized and well-organized bone tissue around the residual biomaterial particles, although with significant resorption over the observation period.

Another study by Canullo et al. [31] employed nano-crystalline HAp as bone filler material for sinus grafting prior to implant placement. After 24 months the mean value of radiographic vertical height of grafted sinus was 13.75 mm, and a 97% success rate of

functionally loaded implants. The authors concluded that nano-crystalline HAp graft could be used in maxillary sinus lift procedures in cases of early loading, absence of membrane on the buccal wall and low residual bone height.

Mertens et al. [32] performed sinus lift procedure using a synthetic, nano-porous bone graft material consisting of a mixture of nano-HAp and nano-β-tricalcium phosphate crystals, combined with blood from the defect side. Implant placement was performed immediately or after 4 months. A vertical bone resorption of 10.32% for the one-stage group and 10.82% for the two-stage group was observed at 21.45 months follow-up. Good primary stability was observed at implant placement. The authors concluded the technique was reliable and predictable.

De Tullio et al. [33] compared the quality of bone formation following grafting of the alveolar socket with (i) calcium sulphate, (ii) nano-HAp, (iii) a combination of calcium sulphate and nano-HAp, or (iv) no biomaterial. At 5 months, the percentage of vital bone was significantly higher in the calcium sulphate plus nano-HAp group (58.72% versus 13.56% for the calcium sulphate and 17.84% for the nano-HAp groups). The control group presented the highest percentage of vital bone (80.68%). The authors concluded that the application of either nano-HAp or calcium sulphate resulted in new bone formation at 5 months, with the combination of the two biomaterials showing a higher percentage of bone formation, although inferior compared with the control group.

Kattimani et al. [34] employed nano-HAp derived from eggshell as a graft material in third molar post-extractive sockets. When comparing nano-HAp and control group (no application of biomaterial), 83.33% and 50% of trabecular bone were found respectively, with earlier bone remodeling and complete bone regeneration being observed in the nano-HAp group.

Khaled et al. [35] compared the outcomes of nano-HAp versus graftless tenting technique with simultaneous implant placement in sinus lift. Bone gain with nano-HAp was significantly higher compared with control group (7.0 mm and 5.0 mm, respectively). Significantly higher bone density and implant stability were also observed for the nano-HAp group.

**Table 3.** Summary of Nano-HAp application in orthodontics.

| Author | Setting, Funding | Study Design | Sample Size Number (% of Females), Mean Age, Number of Sites | Follow-Up (Months) | Interventions | Study Conclusions |
|---|---|---|---|---|---|---|
| Malekpour et al. [26] | Dental Hospital, NR | RCT | 20 patients (60% F), mean age 19.8 years, 40 sites | 4 | 1. Nano-Hap<br>2. No biomaterial<br>3. SofLex disk<br>4. Carbide bur | No significant effect of nano-HAP was seen inreducing tooth color changes after orthodontic debonding |
| Enan & Hammad [27] | Dental Hospital, Public Funding | CT | 20 patients (60% F), age < 20 years, 80 sites | 2 | 1. Glass-ionomer cement 15% Nano-Hap<br>2. Glass-ionomer cement 10% Nano-Hap<br>3. Glass-ionomer cement 5% Nano-Hap<br>4. Glass-ionomer cement | Nano-HAp significantly reduced microleakage around orthodontic bands. |

Abbreviations: CT, controlled trial; RCT, randomized clinical trial; NR, not reported.

**Table 4.** Summary of Nano-HAp application in bone grafts.

| Author | Setting, Funding | Study Design | Sample Size Number (% of Females), Mean Age, Number of Sites | Follow-Up (Months) | Interventions | Study Conclusions |
|---|---|---|---|---|---|---|
| Strietzel et al. [28] | Dental Hospital, NR | CT | 14 patients (50% F), mean age 53.5 years, 14 sites | 7 | Nanocrystalline HAp (Ostims) material | Nano-HAp promoted alveolar ridgewidth gain |
| Canullo & Sisti [29] | Private practice, Self-supported | CT | 20 patients (55% F), mean age 59 years, 42 sites | 24 | Nano-structured Mg-enriched HAp (Mg-e HAP) | Nanostructured Mg-e HAp combined with a Gore-Tex titanium reinforced membrane was effective in vertical ridge augmentation |
| Canullo et al. [30] | Dental Hospital, Self-supported | CT | 20 patients (F not reported), mean age not reported, 20 sites | 12 | Nano-structured Mg-enriched HAp (Mg-e HAP) | Mg-e HA allowed complete healing but presented significant resorption during the experimental time frame |

**Table 4.** *Cont.*

| Author | Setting, Funding | Study Design | Sample Size Number (% of Females), Mean Age, Number of Sites | Follow-Up (Months) | Interventions | Study Conclusions |
|---|---|---|---|---|---|---|
| Canullo et al. [31] | Dental Hospital, Self-supported | CT | 30 patients, (53% F), mean age 58.3 years, 67 sites | 24 | Nanocrystalline HAp and nano-porous silica gel matrix | Nano-HAp was effective in early loading, absence of membrane on the buccal wall and low residual bone height in maxillary sinus lift procedures. |
| Mertens et al. [32] | Dental Hospital, Industry | CT | 66 patients (57% F), mean age 55.26 years, 94 sites | 21.45 | Nano-HAp and nano-β-tricalciumphosphate crystals, combined with blood from the defect side | The biomaterial showed good osseointegration and radiographic bone stability |
| De Tullio et al. [33] | Dental Hospital, NR | RCT | 10 patients (50% F), mean age 47.47 years, 16 sites) | 5 | 1. Nano-HAp 2. Calcium Sulphate 3. Calcium Sulphate plus Nano-Hap 4. No biomaterial | Nano-HAp and CS, alone or in combination in post-extraction sockets, promoted bone formation at 5 months |
| Kattimani et al. [34] | Dental Hospital, NR | RCT | 12 patients (58% F), mean age 25 years, 24 sites | 6 | 1. Nano-Hap 2. No biomaterial | Nano-HAp enhanced bone regeneration in comparison with the control group at early bone modulation phases |
| Khaled et al. [35] | Dental Hospital, Public Funding | RCT | 19 patients (36% F), mean age 41.5 years, 20 sites | 6 | 1. Sinus membrane elevation and augmentation with nano-HAp bone substitute 2. Graftless sinus membrane elevation (tenting technique) with simultaneous implant placement | Nano-HAp bone graft offered superior results in terms of the bone height gain and the relativebone density values compared to graftless tenting technique |
| Al-Ahmady et al. [36] | Dental Hospital, NR | CT | 20 patients (60% F), mean age 8–15 years, 20 sites | 12 | 1. Autologous bone marrow mononuclear cells seeded on a collagen sponge in combination with NANO-HAp and PRF 2. Alveolar bone grafting with iliac crest bone | Autologous bone marrow mononuclear cells in combination with autologous PRF and Nano-HAP was an alternative therapeutic option for alveolar bone cleft |

Abbreviations: CT, controlled trial; RCT, randomized clinical trial; NR, not reported.

Al-Ahmady et al. [36] compared a combination of bone marrow mononuclear cells, autologous PRF, and nano-HAp versus iliac crest bone graft for unilateral alveolar cleft defects. The authors reported 90% complete alveolar bone union in the experimental group, along with reduced donor site complications, faster and better soft tissue healing, and less postoperative pain during the 12 months observation period.

Results are summarized in Table 4.

### 3.2.5. Periodontology and Implantology

Heinz et al. [37] applied nano-HAp paste to the treatment of intrabony defects associated with papilla preservation flap surgical procedure. At 6 months follow-up, the probing bone gain was $4.3 \pm 1.4$ mm in the nano-HAp group and $2.6 \pm 1.4$ mm after access flap surgery alone. Probing Pocket Depth (PPD) significantly decreased from $8.3 \pm 1.2$ mm to $4.0 \pm 1.1$ mm in the nano-HAp group compared with the control group (from $7.9 \pm 1.2$ mm to $5.0 \pm 1.2$ mm).

Jain et al. [38] compared intrabony defects filling with nano-HAp and β-tricalcium phosphate in terms of PPD reduction, Clinical Attachment Level (CAL) gain, gingival recession and radiographic defect depth. Statistically significant reduction in PPD was observed in the nano-HAp group at 3 months, while at 6 months no difference was observed. CAL gain improved in both groups at 3 and 6 months, in the absence of significant difference between groups at the two timepoints. No difference was found in gingival recession, which increased in both groups. Higher radiographic bone fill was observed in the nano-HAp group compared to β-tricalcium phosphate group at 3 months, while no statistically significant difference was seen at 6 months.

Gamal & Iacono [39] evaluated the degree of nano-HAp particles retention on root surfaces in periodontal defects following (i) application of 10–100 nm nano-HAp; (ii) application of 10–100 nm nano-HAp following 24% EDTA gel treatment; (iii) composite graft consisting of equal volumes of 10–100 nm nano-HAp and 63–150 nm β-tricalcium phosphate; (iv) composite graft consisting of equal volumes of 10–100 nm nano-HAp and 63–150 nm β-tricalcium phosphate following 24% EDTA gel treatment. The authors found a higher nano-HAp retention in the presence of EDTA surface treatment and association with β-tricalcium phosphate.

De Wilde et al. [40] compared titanium mini-implants and nano-HAp-coated mini-implants in the rehabilitation of partial edentulism. The authors found a similar immunological response to the different implant surfaces when evaluating IL-6, TGF-β2, MMP-8, CCL-3, IL-8 and IL-1β. No statistically significant difference was found at histomorphometric evaluation with regards to number of inflammatory cells around the two mini-implant surfaces.

Results are summarized in Table 5.

### 3.3. Risk of Bias in Interventional Studies

Results of risk of bias analysis are reported in Table 6. None of the included studies was judged at low risk of bias for all domains. Twenty-four studies [15–18,20–24,27–40] were judged at high risk, and three studies [19,25,26] were assigned a moderate risk of bias.

**Table 5.** Summary of Nano-HAp application in periodontology and implantology.

| Author | Setting, Funding | Study Design | Sample Size Number (% of Females), Mean Age, Number of Sites | Follow-Up (Months) | Interventions | Study Conclusions |
|---|---|---|---|---|---|---|
| Heinz et al. [37] | Dental Hospital, NR | RCT | 15 patients (53% F), age 38–50 years, 28 sites | 6 | 1. Nano-HAp paste plus papilla preservation flap 2. Papilla preservation flap | At 6 months, Nano-HAp application enhanced clinical results for probing bone gain and probing pocket depth reductionscompared with papilla preservation flaps alone. Nano-HAp showed greater pocketreduction, gain in clinical attachment level and defect fill at 3 months |
| Jain et al. [38] | Dental Hospital, Public Funding | CT | 12 patients (50% F), age 20–50 years, 24 sites | 6 | 1. Synthetic nano-sized HAp bone graft 2. β-tricalcium phosphate | |
| Gamal & Iacono [39] | Dental Hospital, Self-supported | RCT | 60 patients (36% F), mean age 36.8 years, 120 sites | 0, 5 | 1. Nano-HAp (10 to 100 nm) 2. 24% EDTA gel plus Nano-HAp (10 to 100 nm) 3. Nano-HAp (10 to 100 nm) and β-tricalcium phosphate (63–150 nm) 4. 24% EDTA gel plus Nano-HAp (10 to 100 nm) and β-tricalcium phosphate (63–150 nm) | Microsized β–TCP was a suitable delivery vehicle for nHA, minimizing its dissemination, especially on surfaces treated with EDTA. |
| De Wilde et al. [40] | Dental Hospital, NR | RCT | 13 patients (46% F), mean age 58.8 years, 25 sites | 2 | 1. Nano-Hap coated mini-implants 2. Titanium mini-implants | Nano-hydroxyapatite-coated surfaces in the transmucosal region yielded similar inflammatory response |

Abbreviations: CT, controlled trial; RCT, randomized clinical trial; NR, not reported.

**Table 6.** Quality analysis of the included studies.

| Authors | Randomization | Allocation Concealment | Operators Blinding | Missing Outcome Data Reported | Missing Outcomes ere Balanced Among Groups | Reasons For Dropout Clearly Specified | Selective Outcome Reporting | Therapist Experience | Statistical Method | Sample Size Estimation | Examiner Calibration |
|---|---|---|---|---|---|---|---|---|---|---|---|
| Al-Ahmady et al., 2018 | yellow | yellow | white | red | green | yellow | green | yellow | red | red | red |
| Anand et al., 2017 | green | green | yellow | yellow | green | yellow | green | yellow | green | green | red |
| Bevilacqua et al., 2016 | green | green | yellow | green | green | green | green | yellow | green | red | red |
| Browning et al., 2012 | red | red | green | yellow | yellow | yellow | green | green | green | green | yellow |
| Canullo et al., 2012 | white | white | white | red | yellow | yellow | green | yellow | green | red | yellow |
| Canullo & Sisti 2010 | white | white | white | red | yellow | green | green | yellow | green | red | green |
| Canullo et al., 2016 | green | green | white | red | yellow | green | green | green | green | green | yellow |
| De Tullio et al., 2019 | red | red | red | red | green | red | green | green | green | green | yellow |
| De Wilde et al., 2015 | yellow | yellow | red | green | yellow | green | green | yellow | green | red | red |
| Enan & Hammad 2013 | yellow | yellow | white | red | red | red | red | green | green | red | green |
| Gamal & Iacono 2013 | green | yellow | red | red | red | green | green | yellow | green | red | green |
| Gopinath et al., 2015 | yellow | yellow | green | red | red | red | green | yellow | yellow | red | red |
| Grocholewicz et al., 2020 | green | yellow | green | red | red | red | green | green | green | green | red |
| Heinz et al., 2010 | green | green | red | red | red | green | green | green | green | green | green |
| Jain et al., 2014 | white | white | white | red | red | green | red | yellow | yellow | red | red |
| Jena et al., 2015 | green | green | green | green | green | green | green | yellow | green | red | red |
| Kattimani et al., 2019 | green | green | green | green | yellow | green | green | yellow | green | red | yellow |
| Khaled et al., 2019 | green | green | red | green | yellow | green | green | yellow | green | red | red |
| Malekpour et al., 2022 | green | green | green | yellow | yellow | yellow | green | yellow | green | green | green |

**Table 6.** *Cont.*

| Authors | Randomization | Allocation Concealment | Operators Blinding | Missing Outcome Data Reported | Missing Outcomes ere Balanced Among Groups | Reasons For Dropout Clearly Specified | Selective Outcome Reporting | Therapist Experience | Statistical Method | Sample Size Estimation | Examiner Calibration |
|---|---|---|---|---|---|---|---|---|---|---|---|
| Mertens et al., 2014 | WHITE | WHITE | WHITE | YELLOW | YELLOW | YELLOW | GREEN | YELLOW | GREEN | RED | RED |
| Najibfard et al., 2011 | YELLOW | YELLOW | YELLOW | YELLOW | YELLOW | YELLOW | YELLOW | YELLOW | GREEN | GREEN | RED |
| Strietzel et al., 2007 | WHITE | WHITE | WHITE | YELLOW | YELLOW | YELLOW | GREEN | YELLOW | YELLOW | RED | RED |
| Vano et al., 2015 | GREEN | GREEN | GREEN | YELLOW | YELLOW | YELLOW | GREEN | YELLOW | GREEN | RED | GREEN |
| Vano et al., 2018 | GREEN | GREEN | GREEN | GREEN | GREEN | GREEN | GREEN | YELLOW | GREEN | GREEN | GREEN |
| Wang et al., 2016 | GREEN | YELLOW | GREEN | GREEN | GREEN | GREEN | GREEN | YELLOW | GREEN | GREEN | YELLOW |
| Wierichs et al., 2020 | GREEN | GREEN | YELLOW | GREEN | GREEN | GREEN | GREEN | YELLOW | GREEN | GREEN | RED |

Color coding. WHITE: not applicable; YELLOW: unclear; GREEN: adequate; RED: inadequate.

## 4. Discussion

In the recent years, the clinical applications of nano-HAp in dentistry have progressively increased, although with non-univocal results. The application of different nano-HAp formulations has been suggested to promote enamel remineralization and prevent caries reversal [15–17]. However, for nano-HAp toothpaste, the outcome observed was similar to the application of fluoride-based toothpaste [15,16]. Nevertheless, it should be noted that applying a fluoride-free toothpaste may be beneficial in reinforcing tooth structure while reducing the risk of fluorosis [15]. According to a Cochrane systematic review on fluoride toothpaste use in children, prevention of tooth decay is proportional to the fluoride concentration employed, although the choice of fluoride toothpaste for young children should be balanced against the risk of fluorosis [41]. Although to date a precise cut-off for fluorosis development is not available, a fluoride intake between 0.05 and 0.07 mg of fluoride per kg of body weight has been suggested [41]. Therefore, it may be concluded that nano-HAp based toothpastes can be safely used as an adjunctive means for caries prevention, in the absence of risks of fluorosis development. The combination with other bioactive materials, such as composites releasing remineralizing substances for the prevention of secondary caries, should also be investigated [42]. Grocholewicz et al. [17] employed nano-HAp in combination with ozone therapy. Ozone therapy has been reported for the management of incipient caries, although at present there is no reliable evidence that the application of ozone gas to the surface of decayed teeth stops or reverses the decay process. [43–45] It can be thus concluded that there is insufficient evidence to recommend a combined treatment with nano-HAp gel and ozone therapy.

Dentin hypersensitivity is the condition where the use of nano-HAp has been most extensively investigated. Hu et al. [46] conducted a comprehensive systematic review on different desensitizing toothpastes, and listed toothpastes potassium, stannous fluoride, potassium and strontium, potassium and stannous fluoride, calcium sodium phosphosilicate, arginine, and nano-hydroxyapatite as effective agents in relieving the symptoms of dentin hypersensitivity. de Melo Alencar et al. [47] specifically focused on nano-HAp application for the relief of dentin hypersensitivity, and concluded that nano-HAp at-home and in-office treatments effectively reduced hypersensitivity compared to other desensitizing agents or placebo/negative controls. Among the nine studies included in the present review reporting on dentin hypersensitivity, two studies [18,23] (reported more favorable outcomes in the nano-HAp treatment groups, two studies [19,25] highlighted a superiority of nano-HAp especially in the immediate post-treatment period, while the remaining five studies [20–24] did not highlight a superiority of nano-HAp treatment over other desensitizing agents. In the light of the present evidence, it can be suggested that nano-HAp could be effective in reducing dentin hypersensitivity, providing non superior efficacy compared with other desensitizing agents.

Application of nano-HAp in orthodontics was reported in two studies [26,27]. Nano-HAp was found effective in reducing micro-leakage under orthodontic bands,26 while no additional effect was noted in reducing tooth discoloration following debonding [26]. The present evidence appears extremely limited, and currently hinders the possibility to draw firm conclusions on the application of nano-HAp in orthodontics. Further evaluation through RCTs is advised.

Nano-HAp application as graft material inducing bone regeneration and incorporation with drugs or bioactive molecules has been extensively discussed [8]. The possibility of nano-HAp to enhance regeneration process has been reported for different purposes. Two studies [28,29] employed nano-HAp for ridge augmentation, where the authors observed overall dimensional stability of the grafted material, and success in implant placement. Another application of nano-HAp is as filling material for sinus lift [25,31,32]. Limited bone resorption with high bone density and implant stability was reported, even in cases of early loading. Although limited, the present evidence suggests a potential role of nano-HAp in sinus lift procedures.

Finally, nano-HAp was employed as graft material in post-extractive alveolar sockets [30,33,34]. Current literature provides contrasting results, as according to de Tullio et al. [33] in the absence of graft material the percentage of vital bone at 5 months was higher than in cases of nano-HAp or calcium sulphate application. Conversely, two studies [30,34] reported improved alveolar socket healing following the application of nano-HAp, with a higher percentage of mineralized bone and earlier healing. Being the evidence insufficient, further research on nano-HAp in post-extractive alveolar sockets is advised.

Two studies [37,38] applied nano-HAp to the treatment of intrabony defects, and consistently reported an improvement in PPD and CAL at three months. One study [39] evaluated nano-HAp retention following root surface treatment with EDTA, and thus suggested root preparation prior to nanomaterial application. One study [40] evaluated the use of nano-HAp for implant coating, finding no difference in inflammatory parameters and immunological response compared to titanium mini-implants. The authors suggested that the nanostructure did not enhance tissue integration due to rapid degradation of nano-HAp and rapid adsorption. However, a systematic review by Bral et al. [48] suggested that nano-HAp particles could enhance osteoblast proliferation, adhesion, and calcium deposition, while improving the bio-functionality of the implant. Nevertheless, the effectiveness of implant coating with nano-HAp requires further investigation.

The present study has some limitations. No meta-analysis could be performed due to the extreme variability in the studies included. Quality analysis highlighted a moderate to high risk of bias, and in particular operators experience and missing data reporting were judged unclear in the majority of studies. All these limitations concur to limit the reliability of the analysis performed. Nevertheless, a growing interest towards the applications of nano-HAp can be observed, highlighting the need for further RCTs on the various applications of this nanomaterial.

## 5. Conclusions

In conclusion, nano-HAp appears to have a great potential in dentistry, with a wide range of applications. The biocompatibility and versatility of nano-HAp encourage the performance of further clinical research to assess the potential of this nanomaterial.

**Author Contributions:** Conceptualization, R.I. and M.N.; methodology, S.G.; software, S.G.; validation, F.G. and M.M.; formal analysis, R.I., M.N. and S.G.; investigation, R.I., M.N. and S.G.; resources, M.R.G.; data curation, R.I. and S.G.; writing—original draft preparation, R.I., S.G. and M.N.; writing—review and editing, F.G., M.M. and M.R.G.; visualization, M.N., F.G., M.M. and M.R.G.; supervision, M.R.G.; project administration, M.R.G.; funding acquisition, M.R.G. All authors have read and agreed to the published version of the manuscript.

**Funding:** This research received no external funding.

**Institutional Review Board Statement:** Not applicable.

**Informed Consent Statement:** Not applicable.

**Data Availability Statement:** Data is contained within the article.

**Conflicts of Interest:** The authors declare no conflict of interest.

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
