# Peer review of "Clinical Applications of Nano-Hydroxyapatite in Dentistry"

_applsci, doi:10.3390/app122110762_

Round 1
Reviewer 1 Report
The submitted paper is a review article on current applications of nanoHA in dentistry. The authors have explored different fields ranging from periodontology and bone regeneration to prevention and caries remineralization.
Although the authors' will to conduct a systematic review, actually they rather performed a narrative review. A systematic review should have focused on one single application. No comparison was done between the included articles and no practical nor interesting conclusions could be drawn from the review.
Table 1 is not suitable. Different tables should be reported according to each application. Conclusions should be briefly reported in the table instead of cut and paste reporting.
Given the number of application of nano-HA in dentistry, the introduction is insufficient.
References are not written according to the style of the Journal.
Author Response
We would like to thank the reviewer for the comments. The manuscript was edited according to reviewer's suggestions. Please find below pint-by-point response.
Reviewer 1
The submitted paper is a review article on current applications of nanoHA in dentistry. The authors have explored different fields ranging from periodontology and bone regeneration to prevention and caries remineralization.
Although the authors' will to conduct a systematic review, actually they rather performed a narrative review. A systematic review should have focused on one single application. No comparison was done between the included articles and no practical nor interesting conclusions could be drawn from the review.
We would like to thank the reviewer for this comment. The type of manuscript was edited accordingly, and the article classified as review.
Table 1 is not suitable. Different tables should be reported according to each application. Conclusions should be briefly reported in the table instead of cut and paste reporting.
Multiple tables were added according to the application of nano-HAp reported. The conclusions were edited as requested.
Given the number of application of nano-HA in dentistry, the introduction is insufficient.
The introduction section was implemented.
References are not written according to the style of the Journal.
Reference style was updated.
Reviewer 2 Report
I appreciated these study wich confirm the biocompatibility and versatility of nano-HAp and the use in many field s of dentistry
Author Response
We would like to thank the reviewer for the comments. The manuscript was edited according to reviewer's suggestions. Please find below pint-by-point response.
I appreciated these study which confirm the biocompatibility and versatility of nano-HAp and the use in many field s of dentistry
We would like to thank the reviewer for this positive remark on our manuscript.
Reviewer 3 Report
Very well presented systematic review. Nano-hydroxyapatite is very promising material in dentistry and its use is described efficiently in this systematic review.
Author Response
We would like to thank the reviewer for the comments.
The manuscript was edited according to reviewer's suggestions. Please find below pint-by-point response.
Very well presented systematic review. Nano-hydroxyapatite is very promising material in dentistry and its use is described efficiently in this systematic review.
We would like to thank the reviewer for this positive remark on our manuscript.
Reviewer 4 Report
Dear authors,
Thanks for submitting your manuscript entitled "Clinical applications of nano-hydroxyapatite in dentistry: a systematic review". My review is below:
-Line 30, please mention exactly the amount or percentage of the mineral in enamel and dentin.
-Your introduction is very short please develop it more.
-I am not sure if you fully finished your introduction because the last word has two points (..)
-Line 37 please mention the sizes of the crystals.
-Line 40 you mentioned that Nano-Hap finds “extensive” application in dentistry but you only provided 2: implantology and tissue regeneration. If it is extensive then mention many more or if not then change the word.
-Line 76, please explain what is unclear abstracts or provide examples.
-For Table 1, please include a column with the title of the publications so it will easier for readers to find the manuscript online.
-Line 272 please mention the concentration causing the risks of fluorosis development
-Before the conclusions, please write a summary of the limitations of the articles you review as an overall and your comments about those limitations and how further studies should be done in order to provide better data.
Author Response
We would like to thank the reviewer for the comments.
The manuscript was edited according to reviewer's suggestions. Please find below point-by-point response.
Dear authors,
Thanks for submitting your manuscript entitled "Clinical applications of nano-hydroxyapatite in dentistry: a systematic review". My review is below:
-Line 30, please mention exactly the amount or percentage of the mineral in enamel and dentin.
The following sentence was added:
In dental enamel, the inorganic matrix accounts for 96% in weight, with HAp crystallites size ranging between 48-78 nm. In mature dentin, the inorganic matrix represents the 70% in weight, and HAp crystallites size ranges between 60–70 nm.
-Your introduction is very short please develop it more.
The introduction section was implemented.
-I am not sure if you fully finished your introduction because the last word has two points (..)
The presence of two full stops was a typo.
-Line 37 please mention the sizes of the crystals.
The following sentence was added:
While synthesized HAp is characterized by a crystal size of 25.40 nm, nano-HAp crystal size ranges between 20 to 80 nm
-Line 40 you mentioned that Nano-Hap finds “extensive” application in dentistry but you only provided 2: implantology and tissue regeneration. If it is extensive then mention many more or if not then change the word.
The following paragraph was added:
Nano-HAp finds several applications in dentistry, due to its characteristics of biocompatibility and bioactivity. Among its properties, the ability to promote re-mineralization appears extremely valuable in presence of early caries, where nano-HAp promotes direct replacement of lost minerals or carries mineral ions to the collagen network. [10] In cases of dentin hypersensitivity, nano-HAp has been reported to induce occlusion of dental tubules by acting as a mineralizing agent. Moreover, nano-HAp is employed for implant coating as it improves bone to implant contact and promotes bone adhesion, along with having bacteriostatic properties. Such characteristics account for the favourable outcomes of nano-HAp use as graft material, enhancing angiogenesis and bone healing.
-Line 76, please explain what is unclear abstracts or provide examples.
The following sentence was added:
(e.g. incomplete reporting, unclear study methods)
-For Table 1, please include a column with the title of the publications so it will easier for readers to find the manuscript online.
We included the reference number for each article in order to simplify the retrieval of the citation.
-Line 272 please mention the concentration causing the risks of fluorosis development
The following sentence was added:
Although to date a precise cut-off for fluorosis development is not available, a fluoride intake between 0.05 and 0.07 mg of fluoride per kg of body weight has been suggested.
-Before the conclusions, please write a summary of the limitations of the articles you review as an overall and your comments about those limitations and how further studies should be done in order to provide better data.
The following paragraph was added:
The present study has some limitations. No meta-analysis could be performed due to the extreme variability in the studies included. Quality analysis highlighted a moderate to high risk of bias, and in particular operators experience and missing data reporting were judged unclear in the majority of studies. All these limitations concur to limit the reliability of the analysis performed. Nevertheless, a growing interest towards the applications of nano-HAp can be observed, highlighting the need for further RCTs on the various applications of this nanomaterial.
Round 2
Reviewer 1 Report
Dear authors,
You have made significant improvements to the manuscript that is now acceptable for publication.
I suggest to furthe revise English.
Nice work.